

# Genetic inactivation of alpha-synuclein affects embryonic development of dopaminergic neurons of the substantia nigra, but not the ventral tegmental area, in mouse brain

Tatiana V. Tarasova[1], Olga A. Lytkina[1], Valeria V. Goloborshcheva[1], Larisa N. Skuratovskaya[2], Alexandr I. Antohin[3], Ruslan K. Ovchinnikov[1,3] and Michail S. Kukharsky[1]

[1] Laboratory of Genetic Modeling of Neurodegenerative Processes, Institute of Physiologically Active Compounds, Russian Academy of Sciences, Chernogolovka, Russia
[2] Institute of general pathology and pathophysiology, Moscow, Russia
[3] Faculty of Biomedical Science, Pirogov Russian National Research Medical University, Moscow, Russia

Corresponding author
Michail S. Kukharsky,
kukharskym@ipac.ac.ru,
kukharskym@gmail.com

## ABSTRACT

Lesion of the dopaminergic neurons of the nigrostriatal system is a key feature of Parkinson's disease (PD). Alpha-synuclein is a protein that is a major component of Lewy bodies, histopathological hallmarks of PD, and is involved in regulation of dopamine (DA) neurotransmission. Previous studies of knockout mice have shown that inactivation of alpha-synuclein gene can lead to the reduction in number of DA neurons in the substantia nigra (SN). DA neurons of the SN are known to be the most affected in PD patients whereas DA neurons of neighboring ventral tegmental area (VTA) are much less susceptible to degeneration. Here we have studied the dynamics of changes in TH-positive cell numbers in the SN and VTA during a critical period of their embryonic development in alpha-synuclein knockout mice. This precise study of DA neurons during development of the SN revealed that not only is the number of DA neurons reduced by the end of the period of ontogenic selection, but that the way these neurons are formed is altered in alpha-synuclein knockout mice. At the same time, DA neurons in the VTA are not affected. Alpha-synuclein exerts a modulating effect on the formation of DA neurons in the SN and has no effect on the formation of DA neurons in VTA, the structure that is much less susceptible to degeneration in a brain with PD, suggesting a potential role of alpha-synuclein in the development of the population of DA neurons in substantia nigra.

## INTRODUCTION

Alpha-synuclein, a small natively unfolded protein abundantly expressed in vertebrate neurons, is involved in pathogenesis of certain neurodegenerative diseases. Principal members of this group of diseases known as alpha-synucleinopathies include Parkinson's disease, Lewy body dementia and multisystem atrophy, but alpha-synuclein pathology is

often observed in the nervous system of patients with other neurodegenerative diseases (*Goedert, Jakes & Spillantini, 2017*).

Although the normal functions of alpha-synuclein are still not fully understood, there is a growing body of evidence that this protein modulates the efficiency of various mechanisms that are important for neuronal physiology, including synaptic vesicle function and synaptic transmission (*Burre et al., 2010*; *Greten-Harrison et al., 2010*; *Janezic et al., 2013*; *Nemani et al., 2010*; *Ninkina et al., 2012*; *Vargas et al., 2017*), mitochondrial ATP production (*Ludtmann et al., 2016*; *Ryan et al., 2015*), survival (*Gorbatyuk et al., 2010*), and sensitivity to neurotoxins (*Robertson et al., 2004*). The key mechanism of pathology of PD, severe loss of DA neurons of the SN, remains a major focus of research on the way to revealing specific molecular targets for developing the disease-modifying drugs. The main findings describing the various factors that are involved in the development and maturation of DA neurons are summarized by *Luo & Huang (2016)*. It is believed that the period between mouse embryonic days 11.5 and 13.5, when post-mitotic precursors of DA neurons are migrating from the ventricular zone located on the border between midbrain and forebrain to their sites in the SN and VTA where their final differentiation takes place, is a critical period in the process of forming these two neuronal populations. This coincides with a dramatic increase of alpha-synuclein expression in midbrain nuclei (*Abeliovich & Hammond, 2007*). Moreover, depletion of alpha-synuclein from developing neurons, leads to a smaller number of mature DA neurons in the SN of knockout mice (*Garcia-Reitboeck et al., 2013*). The aim of this study was to reveal whether alpha-synuclein could have a different impact on the development of DA neurons in the VTA that are much less affected by neurodegenerative processes in the PD brain. Therefore, the numbers of DA neurons in two close populations, the SN and VTA, were analyzed during embryonic development of alpha-synuclein knockout mice.

## MATERIALS AND METHODS

All animal work was carried out in accordance with the Rules of Good Laboratory Practice in Russian Federation (2016). The Bioethics committee of Institute of Physiologically Active Compounds, Russian Academy of Sciences provided full approval for this research (Approval No. 20 dated 23.06.2017). A line of alpha-synuclein knockout mice on C57Bl6J background that was used in this study has been described previously (*Abeliovich et al., 2000*; *Robertson et al., 2004*). The embryos were fixed in cold 4% paraformaldehyde solution, paraffin-embedded, and serial 8 μm transverse sections through the mesencephalon regions were prepared for immunostaining as described previously (*Robertson et al., 2004*). Dopaminergic neurons were visualised by staining sections with anti-tyrosine hydroxylase (TH) antibodies (mouse monoclonal antibody, clone TH-2, Sigma) diluted 1:1,000. Stereological counting of TH-positive cells through the entire lateral region of the mesencephalon was performed as described previously (*Al-Wandi et al., 2010*; *Robertson et al., 2004*). Statistical analysis was performed using GraphPad Prism software (GraphPad, San Diego, CA, USA). A probability value of $p < 0.05$ was considered statistically significant.
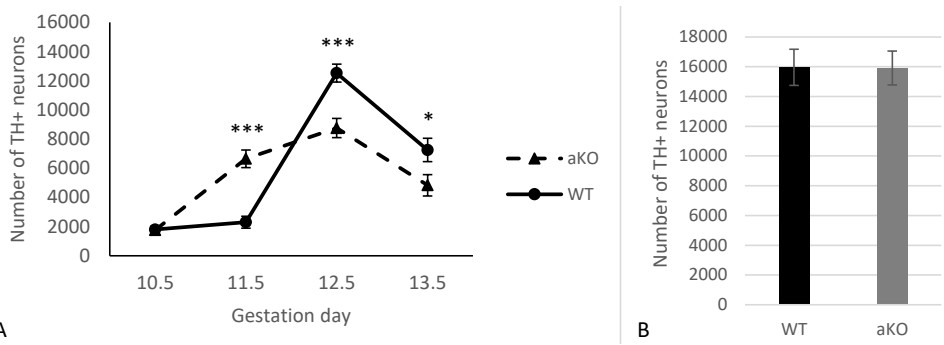

**Figure 1** **The number of TH-positive neurons in SN of alpha-synuclein knockout and control WT mice in embryogenesis (A); the number of TH-positive neurons in VTA, E13.5 (B).** Values represent means ± S.E.M. $n = 5$ for each group and time points. Two-way ANOVA, (*** $p < 0.001$, * $p < 0.05$).

## RESULTS

In this study, we analysed the sub-population of developing dopaminergic neurons within the substantia nigra in alpha-synuclein knockout mice (aKO) and control wild type (WT) littermates on embryonic days E10.5, E11.5, E12.5 and E13.5. The total numbers of TH-positive cells were estimated in the whole SN regions at each embryonic stage. A small number of TH-positive cells in the lateral region of the mesencephalon that represent the primordium of the SN were already detected by embryonic day E10.5 and no difference between knock-out and WT mice was revealed (Fig. 1A and Supplemental Information). In WT mice no substantial increase in the number of these neurons were observed during the next embryonic day, while in aKO mice, a nearly four-fold increase was detected. As a result, the number of TH-positive cells was 2.9 times higher in the SN of E11.5 aKO embryos than in WT embryos at the same stage (Fig. 1A, Fig. 2). A sharp increase in number of TH-positive neurons was observed between E11.5 and E12.5 in WT mice while in aKO mice this increase was less prominent. Whereas aKO mice show an increased number of neurons one day earlier (E11.5), they eventually show a comparatively lower number of TH-positive neurons at all of the following days where measurements were obtained. Therefore, at E12.5, more SN neurons were found in WT than in aKO embryos. At this point the number of TH-positive neurons have reached their maximum number followed by a decline in number on day E13.5 in both conditions. These data clearly demonstrate that in the absence of alpha-synuclein, the dynamics population number in SN neurons is changed.

Additionally, TH-positive cells were counted in VTA on E13.5 day when midbrain dopaminergic neurons are known to separate into SN and VTA zones (*Hu et al., 2004*). The positioning of SN and VTA in embryonic mouse brain for stereological counting of DA neutrons is shown on Fig. 2. As a result, we found that embryonic DA neurons in VTA are not affected in alpha-synuclein knockout mice and their number is the same as in WT control embryonic brains (Fig. 1B).

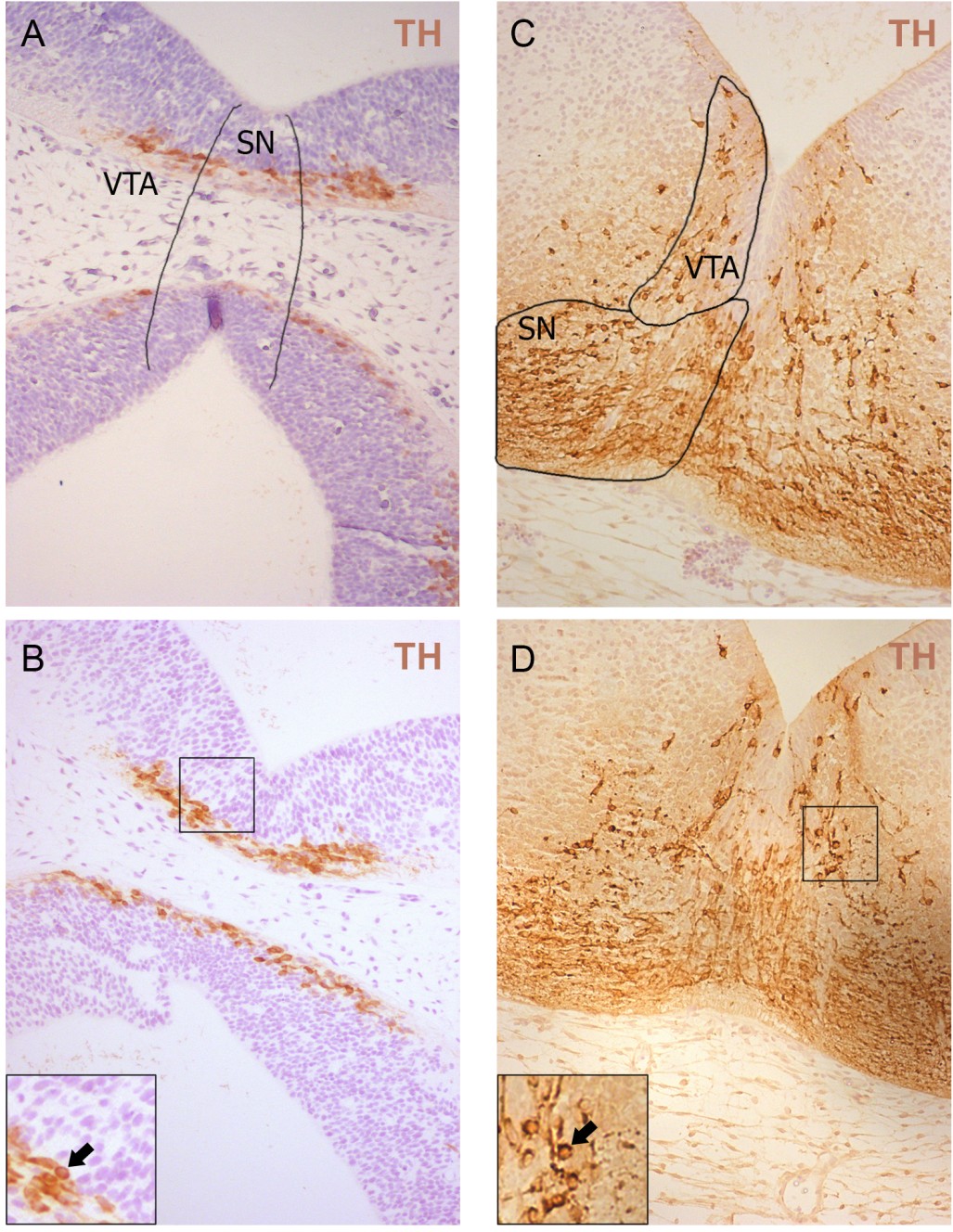

**Figure 2** **Representative developing brain sections from WT (A, C) and aKO (B, D) mice on 11.5 (A, B) and 13.5 (C, D) embryonic days immunostained with anti-tyrosine hydroxylase (TH) antibody.** The areas corresponding to the Substantia Nigra (SN) and Ventral Tegmental Area (VTA) are outlined by the line. TH-positive neurons indicated by arrows at a higher magnification (inserts).

## DISCUSSION

The role of alpha-synuclein in the genesis of dopaminergic neurons of the SN has been investigated in several previous studies (*Garcia-Reitboeck et al., 2013*; *Spillantini & Goedert, 2017*; *Zaltieri et al., 2015*). Loss of function of alpha-synuclein leads to a reduced number of DA neurons in SN at E13.5 (*Garcia-Reitboeck et al., 2013*) and this deficit remains in adulthood (*Robertson et al., 2004*). Here, using littermates as mothers for producing embryos at consecutive stages of embryonic development, we have conducted a precise study of DA neurons in the SN, and have shown that not only the number of E13.5 DA neurons is reduced, but that the way these neurons are formed is changed in alpha-synuclein knockout mice (Fig. 1A). Therefore, alpha-synuclein deficit leads to an inefficient DA neuron pool formation in SN during embryonic development.

It is well established that the degeneration of DA neurons of VTA is less prominent in brains with PD (*Alberico, Cassell & Narayanan, 2015*; *Mosharov et al., 2009*) and that overexpression of the PD mutant alpha-synuclein does not affect VTA (*Maingay et al., 2006*). In our study no changes were found in the number of DA neurons of VTA. These confirm that alpha-synuclein, among other factors, is responsible for the differential sensitivity of SN DA neurons to damage in PD pathology. Further investigation is warranted to determine whether alpha-synuclein deficiency can shape the differential vulnerability of DA neurons to neurodegeneration in the adult and aged brain.

## CONCLUSION

Our data have shown a prominent modulating effect of alpha-synuclein on the formation of DA neurons of the SN and no effect on the formation of DA neurons in VTA suggesting a critical role for this protein in maturation of the population of DA neurons in substantia nigra.

## ACKNOWLEDGEMENTS

Part of the experimental work was performed using the equipment of the Center for collective use of Institute of Physiologically Active Compounds Russian Academy of Sciences.

### Funding

This study was supported by the Russian Foundation for Basic Research (grant No. 16-34-00530), and the Federal Agency for Scientific Organizations program supported the Bioresource Collections No. 0090-2017-0016. The funders had no role in study design, data collection and analysis, decision to publish, or preparation of the manuscript.

### Grant Disclosures

The following grant information was disclosed by the authors:
Russian Foundation for Basic Research: 16-34-00530.
Federal Agency for Scientific Organizations: 0090-2017-0016.

## Competing Interests

The authors declare there are no competing interests.

## Author Contributions

- Tatiana V. Tarasova, Olga A. Lytkina and Valeria V. Goloborshcheva performed the experiments, contributed reagents/materials/analysis tools, authored or reviewed drafts of the paper, approved the final draft.
- Larisa N. Skuratovskaya and Alexandr I. Antohin analyzed the data, authored or reviewed drafts of the paper, approved the final draft.
- Ruslan K. Ovchinnikov performed the experiments, authored or reviewed drafts of the paper, approved the final draft.
- Michail S. Kukharsky conceived and designed the experiments, analyzed the data, contributed reagents/materials/analysis tools, prepared figures and/or tables, authored or reviewed drafts of the paper, approved the final draft.

## Animal Ethics

The following information was supplied relating to ethical approvals (i.e., approving body and any reference numbers):

The Bioethics committee of Institute of Physiologically Active Compounds, Russian Academy of Sciences provided full approval for this research.

## Data Availability

The raw data have been uploaded in a Supplemental File.

## Supplemental Information

Supplemental information for this article can be found online at http://dx.doi.org/10.7717/peerj.4779#supplemental-information.

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
