# Peer review of "Genetic inactivation of alpha-synuclein affects embryonic development of dopaminergic neurons of the substantia nigra, but not the ventral tegmental area, in mouse brain"

_PeerJ, doi:10.7717/peerj.4779_

## Round 0.1 · original submission · Minor Revisions

Please ensure you make all changes suggested by the reviewers (corrections of typos and/or text errors), and improve the figures (including additional wild-type/KO comparisons) as recommended by reviewer 4.

Before resubmitting the manuscript, a careful final proofreading is recommended to exclude any remaining typos or inconsistencies.

Reviewer 1 ·

Basic reporting

Tarasova et al., have studied the dynamic change in TH-positive numbers between E10.5 and E13.5 in the SN of asyn KO mice. They report that asyn KO mice display an earlier increase in the number of TH-positive cells than WT cells. Overall, however, number are lower than in WT mice (agreeing with previous reports, Garcia-Reitboeck P et al., 2013). Moreover, they find no changes in the TH-positive cell numbers in VTA region between the two conditions.

The manuscript is clear and unambiguous, language is fine, references are appropriate.

Experimental design

The experimental design is appropriate, methods are described with sufficient detail and ethical standards have been met (got bioethics approval).

Validity of the findings

Data is robust, statistically sound, & controlled. Conclusion are well stated, linked to original research question & limited to supporting results.

Reviewer 2 ·

Basic reporting

This is a very brief report analysing the ontogenesis of dopaminergic neurons in mutant mice for the alpha-synuclein gene, which has been implicated in the pathogenesis of several neurodegenerative diseases including Parkinson's disease (see below for comments on the experimental procedure and validity of findings).
The manuscript is written in a clear and accessible way.

Experimental design

Up to standards

Validity of the findings

These appear valid

Additional comments

Major comment:

The Y-axis scale in Fig. 1B isn't discriminative enough. No need to modify the figure, but the numerical values (mean+/- s.e.m.) should be added in the legend.


Minor corrections:
In general the manuscript is correctly written and the language is fine. A number of typos or minor errors are found, however, throughout the manuscript. These can be easily corrected:

line 48: replace ; by ,

line 49: add comma (,) after "atrophy"

line 53: "...mechanisms that are important for neuronal physiology; including its roles in synaptic vesicle function..." should read: "...mechanisms that are important for neuronal physiology, including synaptic vesicle function..."

line 58: The key mechanism of pathology of PD

line 62: delete EJ

line 64: ventricular shouldn't be in italics

line 74: add comma (,) between populations and the

line 84: spell PFA

line 90: performed instead of reformed

line 103: delete "In contrast" (sentence should begin with: "In WT mice, no substantial increase..")

line 108 "...while in aKO mice this increase..."

line 128: "leads'

line 134: "...leads to an inefficient DA neuron pool formation..."

line 142 : correct deferential to differential

line 147: "...formation of DA neurons..."



Figure 2 legend: correct Substancia to Substantia

Reviewer 3 ·

Basic reporting

This interesting and carefully executed study shows that the pattern of development of DA neurons in the Substantia Nigra (SN), but not in the Ventral Tegmental Area (VTA) is altered in alpha-synuclein knockout mice. Previous studies in alpha-synuclein knockout mice have shown that inactivation of this gene can lead to the reduction in number of DA neurons in the SN. The brain area known to be the most affected in PD patients. In the present study the dynamics of changes in TH-positive cell numbers in the SN and VTA during a critical period of their embryonic development was evaluated in alpha-synuclein knockout mice.

Experimental design

The authors report that mice deficient in alpha-synuclein have altered number of DA neurons in the SN and have no differences in the formation of DA neurons in VTA, the structure that is much less susceptible to degeneration in PD brain.

Validity of the findings

The manuscript is well written and methods are described with all the necessary details. The observations reported here are novel and provocative.

Additional comments

The only suggestion for authors is to correct in Figure 2 legend – Substantia Nigra.

Reviewer 4 ·

Basic reporting

Sufficient introduction and background context are provided and the relevant prior literature is cited and clearly referenced.

In general, the text is technically correct and conforms to professional standards; however the article does suffer from a number of typographical errors and grammatical mistakes and minor corrections are needed.

The article is structured appropriately and the figures are relevant to the articles’ content. Fig. 1 is clearly described, appears statistically correct (in concert with raw data) and does not present any concerns.

As presented, Fig. 2 provides only two images, one at E11.5 and one at E13.5. These figures are not both labeled, with the VTA and SN shown only in the E13.5 image. No indication is provided whether these are from WT or aKO sections (presumably the former), nor is there clear indication of which neurons are TH+ve.

With regard to Fig. 2 it is suggested that the authors provide additional, clearly labeled images describing WT versus aKO sections at all relevant ages. It should not present a significant challenge to provide this additional data and would greatly strengthen the article and provide further clarity for the reader.

All appropriate raw (statistical) data has been made available in accordance with the PeerJ Data Sharing policy.

Experimental design

All raw data appear to be statistically sound and proper controls have been employed. Imaging data would be greatly improved by addition of side-by-side comparisons of sections from WT and aKO animals. Additional, more detailed labeling of figures is warranted.

Conclusions are concise, appropriately stated and limited only to results and original research question.

Overall, no concerns.

Validity of the findings

All raw data appear to be statistically sound and proper controls have been employed. Imaging data would be greatly improved by addition of side-by-side comparisons of sections from WT and aKO animals. Additional, more detailed labeling of figures is warranted.

Conclusions are concise, appropriately stated and limited only to results and original research question.

Additional comments

Some improvements are suggested, by the inclusion of additional imaging, showing WT vs aKO sections that indicate which neuron populations are labeled as TH+ve.

Some grammatical and typographical errors are present and it would improve the article if these were corrected (a minor concern).

---

## Round 0.2 · Minor Revisions

I am glad to let you know that the manuscript is acceptable, considering the changes made in response to the reviewer's comments. I just noticed two specific points:

- it seems the raw data (numerical values - cell countings) have been made available as a supporting table, which is perfect. However, I couldn't find a reference to this supporting table within the manuscript. It would be good to indicate it wherever necessary.
- Finally, in the Acknowledgments section, you mention the use of collective equipment from "IPAC RAS". It would be useful to spell the name of this center/resource in full.

Sorry for returning to you with these last changes, but I think they will be useful.

---

## Round 0.3 · accepted · Accept

Thanks again for considering PeerJ, and best luck for your future projects.

#